# Acoustical Slot Mode Sensor for the Rapid Coronaviruses Detection

**DOI:** 10.3390/s21051822

**Published:** 2021-03-05

**Authors:** Olga Guliy, Boris Zaitsev, Andrey Teplykh, Sergey Balashov, Alexander Fomin, Sergey Staroverov, Irina Borodina

**Affiliations:** 1Institute of Biochemistry and Physiology of Plants and Microorganisms, Russian Academy of Sciences, Saratov 410049, Russia; guliy_olga@mail.ru (O.G.); strazth87@bk.ru (A.F.); staroverovsergey@hotmail.com (S.S.); 2Kotel’nikov Institute of Radio Engineering and Electronics of RAS, Saratov Branch, Saratov 410019, Russia; zai-boris@yandex.ru (B.Z.); teplykhaa@mail.ru (A.T.); 3Information Technology Center Renato Archer, Campinas CEP, SP 13069-901, Brazil; sergey.balashov@gmail.com

**Keywords:** acoustic slot wave sensor, conducting aqueous solution, coronaviruses, antibodies

## Abstract

A method for the rapid detection of coronaviruses is presented on the example of the transmissible gastroenteritis virus (TGEV) directly in aqueous solutions with different conductivity. An acoustic sensor based on a slot wave in an acoustic delay line was used for the research. The addition of anti-TGEV antibodies (Abs) diluted in an aqueous solution led to a change in the depth and frequency of resonant peaks on the frequency dependence of the insertion loss of the sensor. The difference in the output parameters of the sensor before and after the biological interaction of the TGE virus in solutions with the specific antibodies allows drawing a conclusion about the presence/absence of the studied viruses in the analyzed solution. The possibility for virus detection in aqueous solutions with the conductivity of 1.9–900 μs/cm, as well as in the presence of the foreign viral particles, has been demonstrated. The analysis time did not exceed 10 min.

## 1. Introduction

Viral infections, including coronavirus infection, attract attention with enviable regularity and remain one of the global problems of our time. Coronavirus (Cov) is an acute viral infectious disease characterized by moderate intoxication, and predominantly, affection of the upper respiratory tract. Various types of coronaviruses are widespread in nature, causing various infectious pathologies, both in humans and in animals: pigs, chickens, dogs, cats, camels. Currently, the coronavirus family includes 2 types of viruses that cause severe respiratory infection in humans: SARS-Cov (severe acute respiratory syndrome coronavirus) or SARS coronavirus, which causes severe acute respiratory syndrome, and MERS-Cov (Middle East respiratory syndrome coronavirus) or MERS-coronavirus, causing Middle East respiratory syndrome. At the end of 2019, SARS-Cov-2 (Severe acute respiratory syndrome-related coronavirus 2), capable of causing viral pneumonia [1], was described in Reference [2]. Coronavirus infection is widespread and is recorded throughout the year with peaks in the incidence in winter and early spring, when its epidemic significance ranges from 15.0% to 33.7%. It is important to note that coronavirus infection is zooanthroponous, i.e., transmission of the virus from animals to humans is possible. One of the representatives of coronavirusesis the virus of porcine transmissible gastroenteritis(TGEV), which causes a highly contagious acute infectious disease, mainly in pigs of all age groups, and leads to great economic damage [3,4]. Therefore, the development and improvement of rapid methods for the determination of coronaviruses is very important.

Qualitative analysis of viruses is determined by the reliability of methods for their determination and the ability to study a large number of samples in a short period of time. Various approaches are used to detect viruses (virions), such as microbiological and biochemical tests, genetic engineering and immunological methods [5,6], polymerase chain reaction (PCR) and its various modifications [7,8,9], enzyme-linked immunosorbent assay (ELISA) [10], flow cytometric analysis [11], DNA microarray technology [12,13,14], surface-enhanced Raman scattering (SERS) [15], etc. The current trends in laboratory diagnosis of coronaviruses are shown in Figure 1. With the occurrence of biosensors, traditional approaches to virus detection methods have changed significantly. Biosensor virus detection systems offer unique alternatives to traditional diagnostic tests and provide a low cost, sensitive, fast, miniature and portable platform compared to conventional laboratory methods [16,17,18,19,20].

A promising direction for the analysis of viruses is the use of acoustic biosensors. Previously, the possibility for viruses’ detection was shown on the example of bacteriophages using an acoustic sensor and microbial cells [21], and antibodies [22] and phage mini-antibodies [23] were used as a selective agent. For viruses’ detection, the most preferred sensors are those that allow multiple measurements and are easily cleaned of the used sample after measurements without losing the sensor’s sensitivity to the analyzed reaction. An example is the acoustic sensor based on a slot mode in an acoustic delay line with a zero-order shear-horizontal acoustic wave. The main advantage of this sensor is the possibility of contactless analysis, in which the container with the suspension under study is isolated from the surface of the delay line. This design allows multiple measurements and cleaning of the liquid container without damaging the delay line [24].

The aim of the current work was to develop an express method for detecting viruses using a sensor based on the slot mode in an acoustic delay line. As a model system, we used a representative of the Coronaviride family, transmissible gastroenteritisvirus, and antibodies specific for it (Ab_TGEV_). The general scheme of the experiments is shown in Figure 2.

## 2. Materials and Methods

### 2.1. Virus Samples

For research, the transmissible gastroenteritis (TGE) virus strain VN-96 was obtained from the Museum of Strains of the Virology Laboratory, Saratov Veterinary Research Institute (Saratov, Russia). We used strain VN-96, which was deposited at the All-Russian State Collection of Microbial Strains Used in Veterinary Medicine and Animal Husbandry under accession number VN-96. The main properties of this virus were described earlier in Reference [25].

We used also the bacteriophage M13K07, a kanamycin-resistant commercial preparation manufactured by Stratagene (Sweden), which was constructed on the basis of the wide-type phage M13 [26,27].

The concentration of the viral particles was measured on a Specord BS-250 device (Analytik Jena, Jena, AD, Germany), as described in Reference [22].

### 2.2. Antibodies

Antibodies were obtained by immunizing rabbits with the TGE viruses, and their sensitivity and specificity were determined by the method of dot-immunoassay [22]. The titer of anti-TGEV antibodies (Ab_TGEV_) was determined by the enzyme-linked immunosorbent assay (ELISA), as described in Reference [28], and was 1:6000.

### 2.3. Electron Microscopy

Transmission electron microscopy (TEM) analysis of the TGE viruses was carried out in accordance with Reference [22] by the Libra 120 transmission microscope (Carl Zeiss, Oberkoche, Germany) at an accelerating voltage of 120 kV in the Simbioz Center of the Institute of Biochemistry and Physiology of Plants and Microorganisms, Russian Academy of Sciences (Saratov, Russia).

### 2.4. Solutions

We used a distilled water with conductivity 1.9 μs/cm and a standard phosphate buffered saline with pH 7.0 (g/L): KH_2_PO_4_, 0.43; Na_2_HPO_4_, 1.68; NaCl, 7.2. Its conductivity varied from 4.1 to 900 μs/cm. The solutions conductivity was measured by the HI 8733 conductometer (HANNA, Instruments Inc., Woonsocket, RI, USA).

### 2.5. Description of the Sensor and Measurement Method

To analyze the virus in an aqueous medium, we used a sensor based on the delay line including a piezoelectric plate of lithium niobate of Y-X cut (Y-X LiNbO_3_) with a propagating acoustic wave with shear horizontal polarization of zero order (SH_0_) (Figure 3). The plate thickness was equal to 0.2 mm. These material and wave type were selected as a result of a preliminary theoretical analysis of the piezoactivity of acoustic waves in piezoelectric plates of various orientations. This analysis showed that the SH_0_ wave propagating in the Y-X LiNbO_3_ plate has a significant electromechanical coupling coefficient of 25% for the frequency of 3 MHz and plate thickness of 0.2 mm. Since the shear dimensions of this plate (40 × 20 mm^2^) were significantly larger in comparison with the thickness, the plate was glued along the edges of a rectangular window with dimensions 38 × 18 mm^2^ of a plexiglass holder (see Figure 3). Thus, in this design, the lithium niobate plate was stretched over the holder window and both sides of the plate were mechanically free with maintaining the plane parallelism. On the underside of the plate, two interdigital transducers (IDTs) were pre-applied to excite and receive the acoustic wave. Each transducer had an aperture of 8 mm and contained 5 pairs of fingers with a period of 1.2 mm. The distance between the transducers was 27 mm.

A removable liquid container, the bottom of which was made of a plate of Z-X+30° LiNbO_3_, was positioned above the delay line with the given gap between IDTs. The gap between the container and delay line was provided with 8 μm thick aluminum foil strips. This design of the sensor allowed us to use a removable liquid container, which in turn facilitated cleaning the container from the sample under study and accelerated the analysis of many samples. The constant air gap between the plates ensured repeatability of the results, and also prevented the damage to the delicate delay line during cleaning the container from the spent biological sample. The thickness of the bottom of the liquid container was equal 0.5 mm, the volume of the container was 1.5 mL. Such orientations of the LiNbO_3_ plate were also characterized by a significant value of electromechanical coupling coefficient for SH_0_ wave on the frequency 3 MHz. Thus, the sensor was a structure of two piezoelectric plates separated by an air gap. The sensor characteristics are described in detail in Reference [24].

The analysis was performed using a network analyzer E5071C (Keysight Technologies, Santa Rosa, CA, USA) to which an acoustic sensor was connected. The frequency dependence of the insertion loss of the sensor was measured in the frequency range of 2.6–3.8 MHz. The frequency dependence of the insertion loss of the sensor exhibits resonance absorption peaks associated with the excitation of a slot wave in the structure consisting of two piezoelectric plates with a gap between them [29]. Since the acoustic wave propagating in a thin Y-X LiNbO_3_ plate has a high value of the electromechanical coupling coefficient, the electric field accompanying this wave penetrates in the contacting air. Due to this property, there exists the slot wave propagating in two piezoelectric plates separated by the vacuum (air) gap. The excitation of the slot wave in such structure leads to the appearance of the clearly expressed resonant peaks on the frequency dependence of the insertion loss of the output signal of the delay line. The appearance of the peaks is determined by the fact that the bottom of the liquid sensor is limited in the direction of propagation of the acoustic wave. Each resonant peak corresponds to the case when the width of the second plate is equal to the whole number of the acoustic half-waves. The earlier studies [29] have shown that the depth of these peaks decreases with an increase in the gap between the piezoelectric plates. A gap of 30 μm is the limit for the existence of resonance peaks. Therefore, we have chosen the minimum gap of 8 μm, for which the resonance peaks are well pronounced. With a constant gap between the plates, the depth and frequency of these peaks depends on the velocity and attenuation of the acoustic wave propagating in such a structure. In turn, the wave velocity and attenuation depend on the electrical and mechanical boundary conditions on the surface of the bottom of the liquid container, which borders with the test liquid. A change in the conductivity or viscosity of a liquid leads to the variation of the acoustic wave velocity and attenuation and the output signal of the sensor. It has been previously shown that the specific biological interaction of microbial cells in a liquid suspension with antibodies leads to a change in the conductivity of the cell suspension and, as a consequence, to a change in the frequency and depth of resonant peaks on the frequency dependence of the insertion loss of the sensor [24].

We investigated the possibility of using an acoustic sensor based on the slot mode in an acoustic delay line for detecting viruses in aqueous solutions when they interact with the specific antibodies. Measurements were carried out for TGE viruses’ suspension (10^4^ virus particles/mL), and for various amount of specific Ab_TGEV_ (5, 10, 15, 20, 25 and30μL/mL) added to the viral suspension. We used a distilled water with the conductivity of 1.9 μs/cm and the standard buffer solutions with the conductivity in the range 4.1–900 μs/cm. First, we filled a liquid container with a buffer solution with TGE viruses and measured the sensor readings. Then, anti-TGEV antibodies were added in the suspension and the sensor readings were recorded again. As analytical signal, we used the change in the depth and frequency of the resonant peaks on the frequency dependence of the insertion loss of the sensor after the interaction of viruses with antibodies. The scheme of experiments with an acoustic sensor is shown in Figure 4.

### 2.6. Statistical Analysis

All experiments were performed in 5 replicates, and the final results were calculated from the averaged values. The average error of 5 measurements at each point of the frequency range did not exceed ±2%. The results were statistically processed using standard procedures integrated into Excel 2007 (Microsoft Corp., Redmond, WA, USA). After the arithmetic mean and standard deviation (SD) were found for a given data sample, the standard error of the arithmetic mean and its confidence limits were determined taking into account the Student’s coefficients (n, *p*) (number of measurements n = 5, probability= 95% (*p* = 0.05)). Therefore, all the graphs stated below containing more than 1600 frequency points are given without the error bars, because in the presented scale, they will simply be invisible.

## 3. Results

Coronaviruses belong to the Coronaviridae family and order Nidovirales. This family has 2 subfamilies: Coronavirinae and Torovirinae. Coronavirinae are categorized into 4 groups: (1) Alphacoronaviruses, (2) Betacoronaviruses, (3) Gammacoronaviruses and (4) Deltacoronaviruses [30]. Their name comes from the Latin word “coronam”, because peplomers create a pronounced jagged frame in the form of a “crown” around the virion shell on the electron microscopic image [31]. The causative agent of TGE is an RNA-containing virus belonging to the Coronaviride family and having the genus Alphacoronavirus group 1a. The virion has a spherical shape and a diameter of 75–160 nm. The viral nucleocapsid is a flexible helix containing single-stranded RNA and a large number of nucleocapsid protein molecules [32,33].

The virus multiplies in the cytoplasm of mature epithelial cells located at the tips of the villi of the small intestine. Coronaviruses are infected by droplet–air and fecal–oral routes (see Figure 5). The virus is highly contagious, spreads rapidly among animals and causes great harm to the food and agricultural industries. The prevalence in most European countries is around 100%. Virus strains isolated from animals in different countries are serologically identical, although different variants have appeared in recent years [31]. Since coronaviruses are a family of related viruses, TGE virus was used as a model sample, as a representative of coronavirus infection.

The morphology of viral particles, presence of the aggregates and individual subunits affects to the process of immunochemical interaction, which, in turn, changes the analytical signal. In this regard, at the beginning of the experiments, TGE viruses were investigated by electron microscopy (TEM). Figure 6a shows a TEM image of the TGE virus without specific antibodies. The TEM results confirmed the correspondence of the size of the viruses with the data obtained in References [25,31]. Additionally, TEM was used to analyze the interaction between the TGE viruses and Ab_TGEV_. Electron micrographs presented in Figure 6b show an image of TGEV with Ab_TGEV_. It can be seen that markers appear on the surface of the virus, i.e., it can be argued that anti-TGEV antibodies interact with TGEV and can potentially be used for its immunodetection.

Then, experiments were carried out to detect the TGE viruses in aqueous solutions using an acoustic sensor. At the first stage, the interaction of viruses with different amounts of antibodies in water with a conductivity of 1.9 μs/cm was studied. First, virus-free distilled water was added to the sensor container and the sensor reading was measured.

The frequency dependence of the insertion loss of the sensor showed peaks of resonant absorption (Figure 7a, black curve). Then, the viruses diluted in distilled water (10^4^ virus particles/mL) were placed in a liquid container and the sensor readings were taken again. This concentration of TGE viruses was selected based on the previous experiments. It was found that the addition of the virus to distilled water does not lead to a change in the depth and frequency of the resonant peaks in the frequency dependence of the insertion loss of the sensor (Figure 7a, green curve).

Next, antibodies specific to this virus (Ab_TGEV_) were added to the container and the sensor readings were recorded for this case. The values of the concentration of antibodies were 5, 10, 15, 20, 25 and 30 μL/mL. It was found that the addition of Ab_TGEV_ to TGEV led to a significant decrease in the depth of resonance peaks and to a shift in resonance frequencies for any amounts of antibodies.

Figure 7b shows, as an example, the frequency dependences of the insertion loss of the sensor before (blue curve) and after (pink curve) the addition of Ab_TGEV_ (in the amount of 20μL/mL) to TGEV.

Based on the obtained frequency dependences of the insertion loss of the sensor, the changes in the depth and frequency of the resonant peaks as functions of the concentration of antibodies added to the container with the virus for all resonance peaks were plotted. The dependences of changes in the depth and frequency of the resonance peak near the frequency of 2.82 MHz on the amounts of added antibodies are presented in Figure 8a,b, respectively.

One can see that the maximum change in the depth of the resonance peak is equal to 7.9 dB with the amount of antibodies being 20 μL/mL. At that, the minimum change turned out to be ~6 dB for antibodies in the amounts of 5 and 30 μL/mL. In this case, for all concentrations of antibodies, a shift in the frequency of the resonance peak by 0.01 MHz was observed. Approximately the same results were obtained for other resonance peaks. Table 1 shows the change in the depth of all resonance peaks observed on the frequency dependence of the insertion loss of the sensor.

Since the maximum change in the depth of the resonance peak was recorded when the amount of antibodies was 20 μL/mL, in subsequent experiments, we used exactly this amount of antibodies to register the interaction of TGE viruses with Ab_TGEV_. It should be noted that the addition of only Ab_TGEV_ in liquid container in the virus’ absence did not lead to changes in the sensor parameters.

Thus, it has been shown that the addition of specific antibodies to the TGE virus under conditions of low conductivity of the medium (1.9 μs/cm) leads to a significant change in the depth of the resonance absorption peaks on the frequency dependence of the insertion loss of the acoustic sensor based on the slot mode. This indicates the possibility of using such acoustic sensor to register the TGE viruses–Ab_TGEV_ interaction during virus detection. The measurement time was ~10 min, including 4 min for the measurement process and the time for the introduction of the sample into the liquid container and cleaning the container after measurements.

When developing a new method for virus detection, it is important to take into account the possibility of analyzing the viruses in solutions with increased conductivity. High conductivity creates significant interference in the analysis and most sensor devices are not adapted for measurements in real conditions [34]. Nevertheless, an important problem is the analysis of viruses in water used in industry, municipal and commercial institutions, and hospitals. The specific electrical conductivity of water in these cases, as a rule, is high due to the dissolved ionic compounds (more then 500 μs/cm). Therefore, we further evaluated the sensitivity of the sensor with the liquid samples of increased conductivity. We studied the interaction of TGE viruses with anti-TGEV antibodies in buffer solutions with conductivity of 4.1, 20, 50, 100, 300, 500 and 900 μs/cm. In this case, the procedure for preparing the sample, introducing Abs and carrying out measurements was similar to that described above. It was shown that the addition of Ab_TGEV_ to buffer solutions with virus led to a decrease in the depth of resonance peaks on the frequency dependence of the insertion loss of the sensor and to a shift in the resonance frequency. In this case, the depth of the resonance peaks depended on the conductivity of the buffer solution. Figure 9 shows, as an example, the dependence of the change in the depth of the resonance absorption peak near the frequency of 2.82 MHz on the conductivity of the buffer solutions. It can be seen that the change in the depth of the resonance peak has maximum value of 7.5 dB at the buffer conductivity of 4.1 μs/cm. With an increase in the conductivity, the change in the depth of the resonance peak upon adding antibodies to the buffer solution with the virus decreases, reaches zero at a buffer conductivity of 250 μs/cm, then changes sign, slightly increases, and reaches saturation at the conductivity of 900 μs/cm.

This behavior of the dependence is in full agreement with the data obtained earlier in the study of the effect of liquid conductivity on the propagation of the slot mode in a structure of two piezoelectric plates separated by an air gap [35]. It was found that with an increase in the conductivity of the liquid, the frequency of the resonance peaks at first remains almost constant, then decreases, and after that does not change. In this case, the dependence of the peak depth on the conductivity of the liquid has a maximum at the conductivity of 250 μs/cm, which corresponds to the condition 2πfτ ≈ 1, where f is the wave frequency, and τ = ε/σ is the Maxwell relaxation time (σ = conductivity, ε = dielectric constant). Figure 8 shows that the sign reversal of the change in the depth of the resonance peak is observed precisely at a conductivity value of 250 μs/cm.

Thus, the interaction of viruses with specific antibodies leads to an increase in the conductivity of the suspension. This suspension is in contact with one of the plates of the structure with a piezoactive slot mode accompanied by an electric field in the suspension. A change in the conductivity of a suspension alters the attenuation and phase velocity of the slot mode. This, in turn, changes the depth and frequency of the resonance peaks in the frequency dependence of the insertion loss of the sensor. The depth and frequency of resonance peaks are directly measurable quantities and are therefore used as an analytical signal, indicating the presence of viruses in suspension. Thus, the possibility of detecting the virus in aqueous solutions with increased conductivity, for example, in drinking and tap water, the conductivity of which is 500 and 900 μs/cm respectively, has been shown.

For the development of virus method detection, an important point is to obtain a result in the presence of interfering factors and, above all, in the presence of other viral particles. Therefore, at the next stage, we measured the parameters of the sensor for a mixed suspension of viruses in the container. M13K07 bacteriophages were used as foreign viral particles. The measurements were carried out in the buffer solutions with the conductivity of 4.1, 50 and 100 μs/cm.

First, the control experiments were carried out to study the nonspecific interaction between M13K07 and Ab_TGEV_. For this, the depth and frequency of the resonant peaks on the frequency dependence of the insertion loss of the sensor were measured before and after the addition of antibodies specific to TGE virus to the buffer solution with the M13K07. Figure 10a shows the frequency dependences of the insertion loss of the sensor for a buffer solution with a conductivity of 4.1 μs/cm containing the M13K07 before (black curve) and after (purple curve) the addition of Ab_TGEV_. It can be seen that nonspecific interaction “M13K07–Ab_TGEV_” is absent and did not lead to a change in the sensor parameters.

Then, the same measurements were carried out for buffer solutions with TGE virus and specific Ab_TGEV_. Figure 10b shows the frequency dependences of the insertion loss of the sensor before (blue curve) and after (red curve) the addition of specific antibodies to a solution with a conductivity of 4.1 μs/cm containing the TGE virus. It can be seen that in this case, the depth of the resonance peaks in the frequency dependence has significantly decreased. The frequency shift of each resonance peak by ~0.01 MHz was also observed.

Finally, a mixed suspension of TGE viruses (10^4^ particles/mL) and M13K07 (10^6^ particles/mL), taken in the 1:1 ratio, was introduced into the sensor container. After that, the Ab_TGEV_ (20 μL/mL) was added. The sensor readings were measured before and after adding Ab_TGEV_ to the mixed virus suspension. Figure 10c shows the frequency dependences of the insertion loss of the sensor for a mixed viruses’ suspension (TGE with M13K07) before (green curve) and after (orange curve) the addition of Ab_TGEV_. One can see that the addition of Ab_TGEV_ to a mixture of viruses leads to approximately the same change in the depth and frequency of resonant peaks, as in the case of a TGE virus suspension without M13K07.

For comparison, Table 2 shows the data obtained during registration of the effect of Ab_TGEV_ on M13K07, TGE virus and mixed suspension (TGE virus and M13K07) in the buffer solutions with conductivity of 4.1, 50 and 100 μs/cm.

One can see that for various values of the conductivity of the buffer solution, changes in the depth of the resonant peaks in the cases of Ab_TGEV_ interaction with the TGE virus and a mixed viruses’ suspension (TGEV and M13K07) are comparable and are equal to 2–5 dB, depending on the buffer conductivity. The changes in this parameter after adding Ab_TGEV_ to the buffer solutions with the M13K07 are greatly smaller and are equal to 0.05–0.11 dB, which is within the sensor error. Thus, it can be argued that the presence of foreign viral particles does not interfere with the registration of the specific interaction of Ab_TGEV_ with TGE viruses by using an acoustic sensor.

## 4. Discussion

As a result of the experiments, it has been found that an acoustic sensor based on the slot mode in an acoustic delay line successfully registers the interaction of the specific antibodies with the TGE virus and is promising for TGE virus detection in aqueous solutions with different conductivity, as well as in the presence of foreign viral particles. The interaction of viruses with specific antibodies leads to an increase in the conductivity of the suspension. A change in the conductivity of a suspension alters the attenuation and phase velocity of the slot mode. This, in turn, changes the depth and frequency of the resonance peaks in the frequency dependence of the insertion loss of the sensor. Based on the changes in the output parameters of the sensor for a virus suspension before and after biological interaction with specific antibodies, it is possible to draw conclusions about the presence (or absence) of the studied viruses in the analyzed suspension. So, the change in the depth and frequency of the resonant absorption peaks on the frequency dependence of the insertion loss of the sensor may be used as an analytical signal.

Let us compare the coronavirus detection method presented in the article with the known ones. In modern methods of detecting coronavirus infection, two main areas can be distinguished:-Detection of viral RNA,-Detection of antibodies produced upon contact with an infection [36].

The detection of viral RNA is usually performed by polymerase chain reaction (PCR) or nucleic acid hybridization.

Viral antibodies or antigens can be detected by immunological and serological tests such as ELISA.

Both directions of research are important, and they complement each other, since the determination of the RNA of the virus leads to the detection of the virus in its active stage, while serological tests help to identify the activity of the immune system in the production of antibodies to fight the infection. Exceptional growth in biosensor development for the analysis of viruses, including coronaviruses, is noted in 2019–2020 [37]. Biosensors are an addition and alternative to traditional diagnostic tests, providing an inexpensive, sensitive, fast, miniaturized and portable platform compared to conventional laboratory methods. Biosensors for detecting coronaviruses can be conditionally divided into the following groups:-Sensors for the viral antigen analysis,-Sensors for the specific antibodies’ determination.

Brief information about modern sensors developed for the coronavirus infection diagnosis is presented in Table 3.

As can be seen from the data presented here, with all the variety of sensor systems for diagnosing coronaviruses, acoustic biological sensors based on the slot mode in an acoustic delay line have not been used to solvethis problem. Acoustic sensors for detecting viral particles are known and described in the literature [54,55], but for the analysis of coronaviruses, their use has not been investigated.

The described method fundamentally differs from the known ones for detecting viruses with immobilized antibodies [54,55,56,57] by the absence of the need to immobilize the components of the analysis, the ease of performing the analysis procedure, high sensitivity and speed of obtaining the results. The proposed method is quite efficient, since the analysis time is about 10 min, including the duration of the measurements and the cleaning of the spent material from the container. At that, the measurement process can be fully automated. In the known literature, there is no data on the use of an acoustic sensor based on a slot mode for detecting coronaviruses, therefore, the studies carried out are pioneering. An additional advantage of the sensor is the presence of a removable liquid container, which allows reusing it and facilitates the process of its cleaning from the spent sample. This is an important condition when working with viruses. The developed method for TGE virus detection may be useful for the rapid monitoring of a large amount of material with minimal time costs in veterinary medicine, healthcare and environmental control. This method can be considered as an additional test for quick diagnoses of coronavirus infection on a large number of samples.

The presented setup showed for the first time the ability to quickly detect viruses in the aquatic environment. Without a doubt, it can be modernized and made more convenient for analysis. It consists of a sensor and an expensive S-parameter meter E5071C (Keysight Technologies, Santa Rosa, CA, USA) (frequency range 9 kHz–8.5 GHz). In the future, it is planned to make a cheaper S-parameter meter for the 1–10 MHz range, combined with a computer with software for fast processing of results.

## 5. Conclusions

Thus, an acoustic sensor based on the slot mode in an acoustic delay line provides a unique opportunity for TGE virus detection under conditions of increased conductivity of the measurement medium and the presence of foreign viral particles. The analysis time did not exceed 10 min. The results obtained can serve as a basis for the development of a new direction in the use of acoustic sensors for detecting viruses.

## Figures and Tables

**Figure 1 sensors-21-01822-f001:**
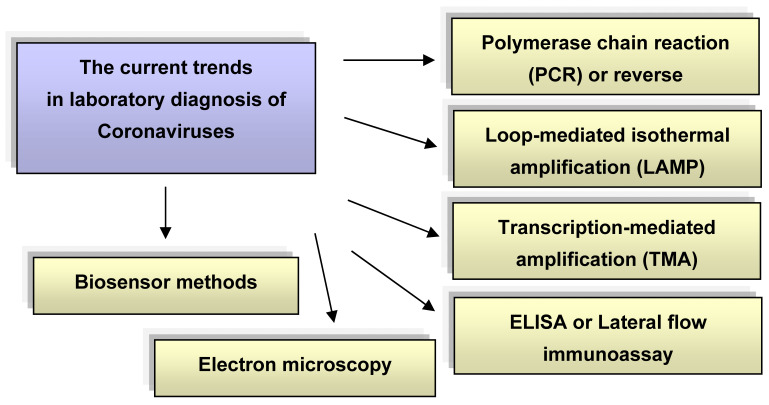
Modern trends in laboratory diagnosis of coronaviruses.

**Figure 2 sensors-21-01822-f002:**
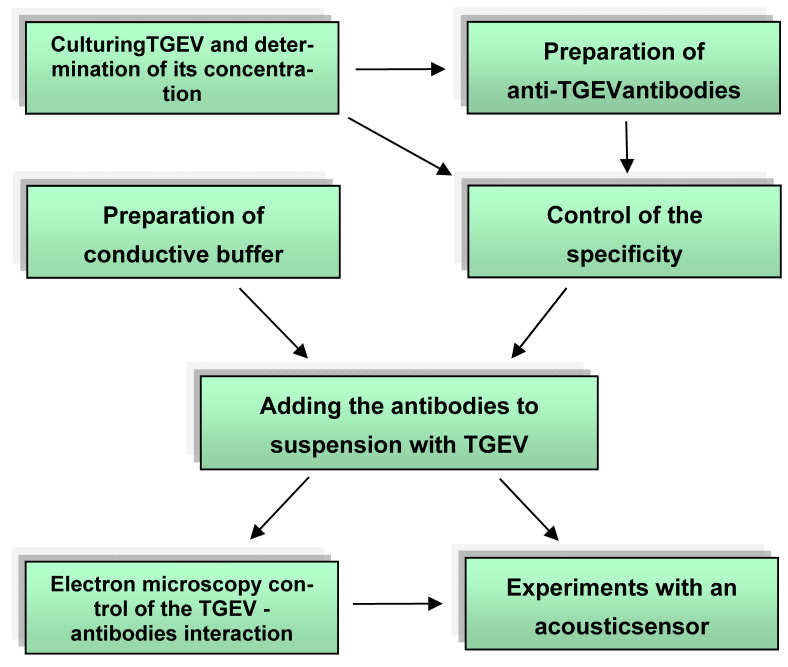
General scheme of the experiment.

**Figure 3 sensors-21-01822-f003:**
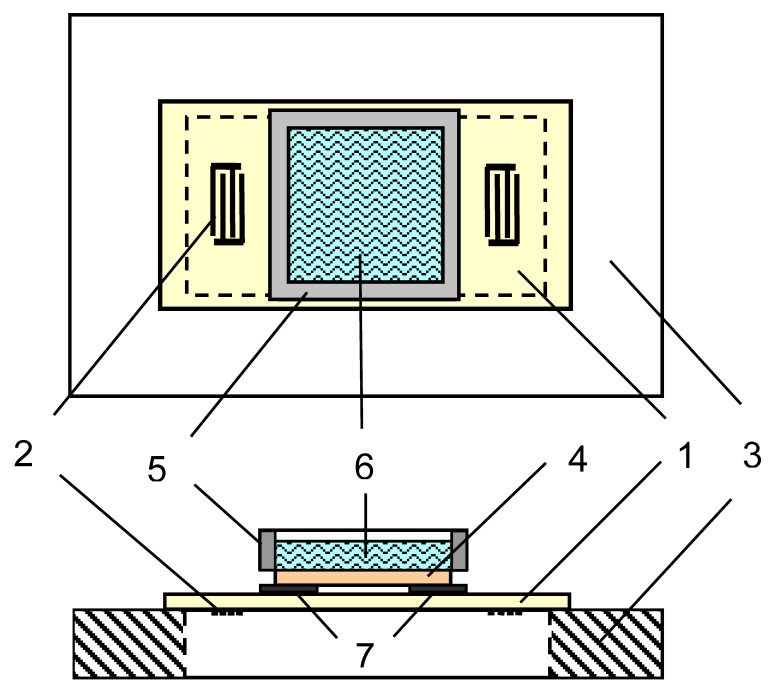
The scheme of the acoustic slot wave sensor: 1: piezoelectric plate of Y-X LiNbO_3_, 2: IDTs, 3: the holder of the plexiglass, 4:piezoelectric plate of Z-X+30° LiNbO_3_, 5:liquid container,6:suspension of the cells under study, 7: the strips of the aluminum foil.

**Figure 4 sensors-21-01822-f004:**
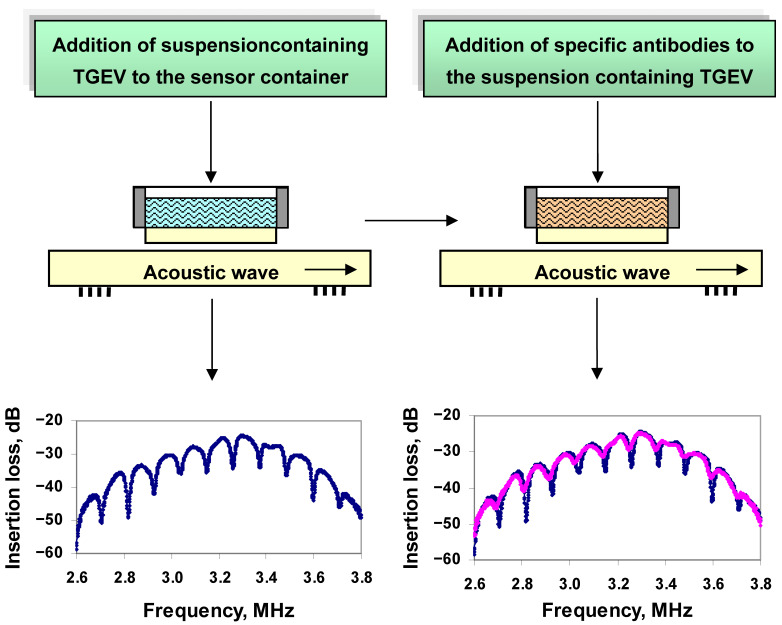
Scheme of the experiments with an acoustic sensor.

**Figure 5 sensors-21-01822-f005:**
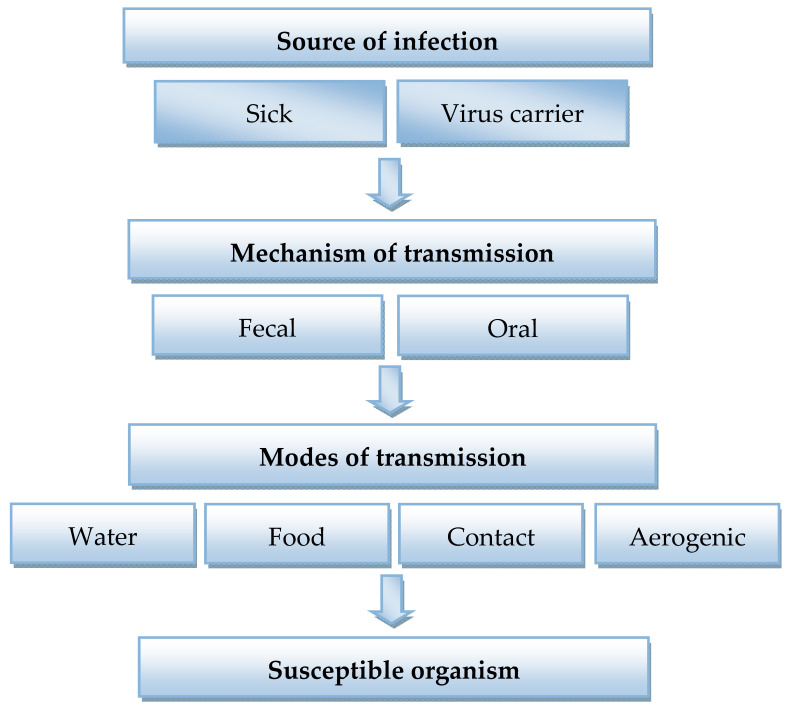
Epidemiology of coronavirus infection.

**Figure 6 sensors-21-01822-f006:**
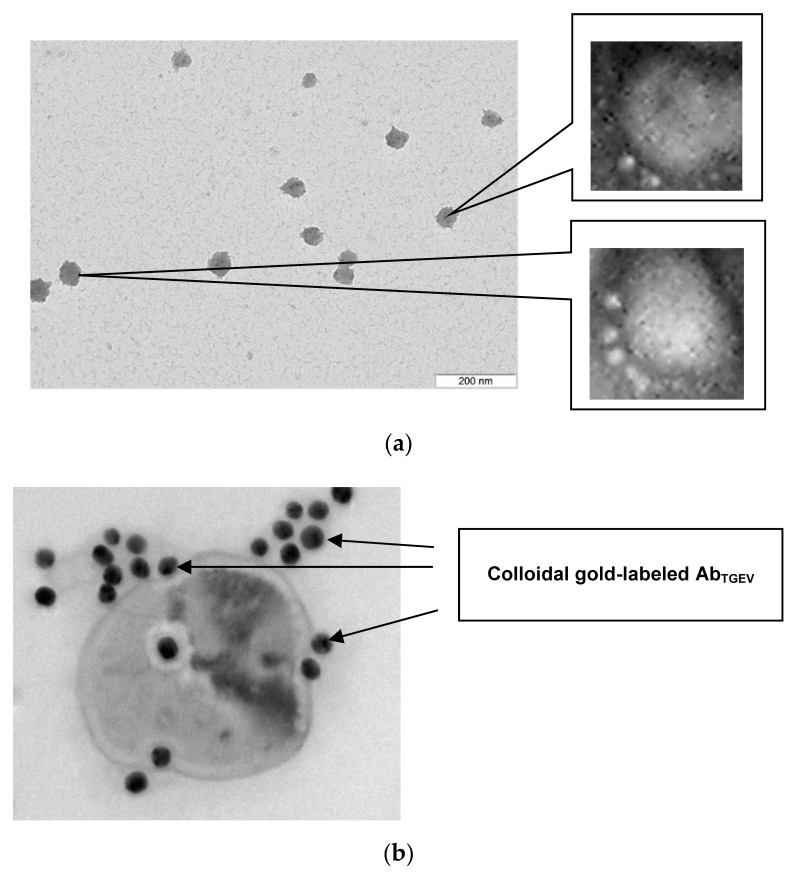
Transmission electron microscopy (TEM) data of TGE viruses: (**a**) without specific Abs and (**b**) with Ab_TGEV_ labeled with colloidal gold (×10,000). The scale is 100 nm (the zoom is equal to 40,000).

**Figure 7 sensors-21-01822-f007:**
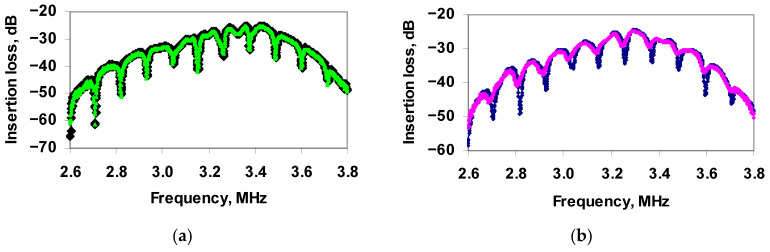
Frequency dependences of the insertion loss of the sensor for distilled water: (**a**) without viruses (black curve) and with TGE viruses (green curve), and (**b**) with TGE viruses before (blue curve) and after (pink curve) adding Ab_TGEV_.

**Figure 8 sensors-21-01822-f008:**
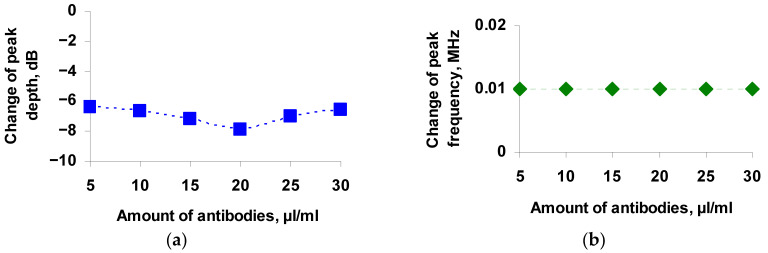
Dependences of the change in the depth (**a**) and frequency (**b**) of the resonance peak near the frequency of 2.82 MHz on the amount of Ab_TGEV_ added to TGE virus.

**Figure 9 sensors-21-01822-f009:**
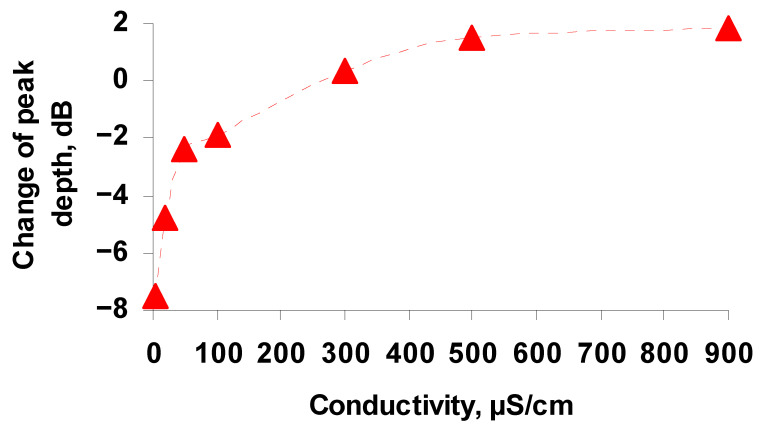
The dependence of the change in the depth of the resonance peak for the suspension with TGE virus upon addition of anti-TGEV antibodies near the frequency of 2.82 MHz on the conductivity of the buffer solution.

**Figure 10 sensors-21-01822-f010:**
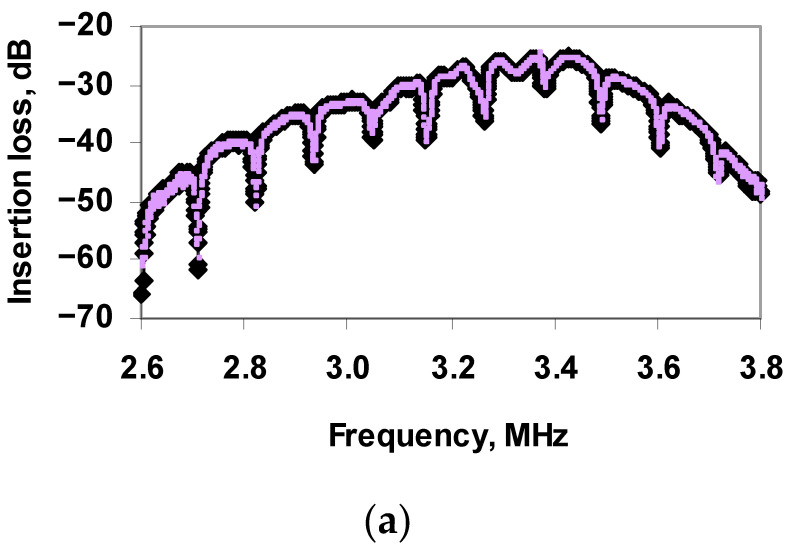
Frequency dependences of the insertion loss of the sensor for buffer solution: (**a**) suspension with M13K07 before (black curve) and after (purple curve) adding Ab_TGEV_, (**b**)suspension with TGE virus before (blue curve) and after (red curve) adding Ab_TGEV_, (**c**) suspension with TGE virus and M13K07 before (green curve) and after (orange curve) adding Ab_TGEV_.

**Table 1 sensors-21-01822-t001:** Change in the depth of all resonance peaks observed on the frequency dependence of the insertion loss of the sensor.

**Amounts of Antibodies, μL/mL**	**Frequency, MHz**									
	2.71	2.82	2.93	3.04	3.15	3.26	3.37	3.48	3.59	3.7
	**Change of the resonance peak depth, dB**									
5	6.2	6.4	7.1	3.7	3.8	5.7	4.1	5.7	5.8	4.1
10	4.46	6.7	5.81	2.81	2.97	4.99	4.91	4.29	3.67	3.86
15	5.3	7.2	5.69	2.8 7	3.81	4.5	5.18	3.86	5.01	4.01
20	6.8	7.9	6.72	3.81	4.87	5.95	5.41	5.81	6.29	4.93
25	5.38	7.05	5.23	2.91	3.29	5.29	2.81	5.23	3.77	3.84
30	6.32	6.59	5.48	3.47	3.51	3.91	4.99	4.78	3.95	4.66

**Table 2 sensors-21-01822-t002:** Change in the depth of the resonant peak near the frequency of 2.82 MHz due to the addition of the antibodies specific to TGE virus in the suspension of M13K07, the suspension of TGE virus, and the mixed suspension (TGE virus and M13K07) at the various values of the conductivity of the buffer solution.

Conductivity of Solution, μs/cm	Change the Depth of the Resonance Peak, dB		
	M13K07 + Ab_TGEV_	TGEV + Ab_TGEV_	TGEV + M13K07 + Ab_TGEV_
4.1	0.11	4.61	5.28
50	0.09	2.37	3.1
100	0.05	1.89	2.2

**Table 3 sensors-21-01822-t003:** Brief information about the variety of biosensors for the coronaviruses analysis.

No	Type of Sensor	Literary Source
1	Dual-function plasmon biosensor using the plasmon photothermal (PPT) effect and localized surface plasmon resonance (LSPR)	[38]
2	Reverse Transcription Loop-Mediated Isothermal Amplification (RT-LAMP) assay	[39,40,41]
3	Nano-biosensor based on the field-effect transistor (FET) method containing antibodies against S-protein loaded on graphene sheet.	[42]
4	Biosensors based on semiconductor field-effect devices (FEDs) enrichment on densely antibody- or aptamer-equipped sensors	[43]
5	The sensor based on the field-effect transistor utilizing the coating of the graphene sheets with a monoclonal antibody against the severe acute respiratory syndrome coronavirus SARS-CoV-2 spike protein. They determined its sensitivity using antigen protein	[44]
6	Biological sensor using Clustered Regularly Interspaced Palindromic Repeats CRISPR-Chip coupled with a graphene-based field-effect transistor	[45]
7	Graphene field-effect transistor (Gr-FET)	[46]
8	Cell-based potentiometric biosensor	[47]
9	Surface plasmon resonance-based colorimetric nano-biosensor containing N-protein loaded on AuNP	[48]
10	Electro-optical virus analysis	[49]
11	Biosensor by using lanthanide-doped polystyrene nanoparticles NPs containing anti-coronavirus -19 Immunoglobuline G COVID-19 IgG based on lateral flow immunoassay	[50]
12	Electrochemical sensor based on the diagnosis of reactive oxygen species (ROS) in fresh sputum in real time using electrochemical tracing and shows the correlation between the determination of viral ROS in the epithelium of the lungs and COVID-19	[51]
13	Electrochemical sensor using AuNPs containing S-protein loaded on carbon electrodes	[52]
14	Piezoelectric immunosensor	[53]
15	Acoustic sensor based on the slot mode in an acoustic delay line	This article

## Data Availability

Not applicable.

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
