# Peer review of "Acoustical Slot Mode Sensor for the Rapid Coronaviruses Detection"

_sensors, 2021, doi:10.3390/s21051822_

Round 1

Reviewer 1 Report

The submitted manuscript suggests using an acoustic slot mode sensor for Coronaviruses detection, and performs an experimental feasibility study with viruses embedded in aqueous solutions with different conductivity. Although the proposed experimental setup seems to have been tested before on a variety of different biological samples, there use as Coronaviruses detectors seems to be novel and also highly relevant. The experimental results also seem to be consistent in view previous finding by the group. However, the manuscript should be improved in terms of clarifying important experimental parameters and physical principles for the used setup. The discussion part should also be improved in terms of suggesting how virus solutions with the specific antibodies, correlates to the measured physical quantities.         

In detail is will suggest implementing the following changes:

  1. The manuscript is quite shallow in describing the experimental setup which makes it very hard to understand the involved physics without reading some of the groups previous publications. This should be improved, e.g. by explaining the underlaying principles for the used slot mode detector.
  2. The role of the narrow airgap between the two LiNbO3 sheets should also be explained. Why is this gap needed and what is done to assure that the tiny gap is kept constant over a relatively large area? It should also be emphasized how the wave energy is transferred between the sheets. I guess the electric field must play an important role here since the mechanical wave coupling through the airgap is quite weak?
  3. The illustration of the experimental setup shown as the middle part of Fig. 3, should be improved by including physical dimensions. Dimensions and volume for the used fluid contained should also be included in the figure and/or manuscript.
  4. From what is said in the manuscript and in discussion, it’s rather difficult to identify the correlation between the observed peek shifts, and the contents of aqueous solutions. I would therefore recommend putting more emphasize on this in the discussion. Correlation to changes in the fluid conductivity is already mentioned, but changes in fluid wave properties (e.g. attenuation and phase velocity) might also be important here?
  5. It would also be interesting if the authors could discuss improvement of the current system with respect to for Coronaviruses detection. Perhaps there is something to gain from miniaturization the setup and increasing the frequency?

Author Response

Answers on the comments of the Reviewer 1

The submitted manuscript suggests using an acoustic slot mode sensor for Coronaviruses detection, and performs an experimental feasibility study with viruses embedded in aqueous solutions with different conductivity. Although the proposed experimental setup seems to have been tested before on a variety of different biological samples, there use as Coronaviruses detectors seems to be novel and also highly relevant. The experimental results also seem to be consistent in view previous finding by the group. However, the manuscript should be improved in terms of clarifying important experimental parameters and physical principles for the used setup. The discussion part should also be improved in terms of suggesting how virus solutions with the specific antibodies, correlates to the measured physical quantities.         

In detail is will suggest implementing the following changes:

  1. The manuscript is quite shallow in describing the experimental setup which makes it very hard to understand the involved physics without reading some of the groups previous publications. This should be improved, e.g. by explaining the underlaying principles for the used slot mode detector.

Answer: Thanks for this comment. We have added Fig. 3 on the page 4 and the following texts on the pages 4 and 5.

Page 4:

“Since the shear dimensions of this plate (40×20 mm2) were significantly larger in comparison with the thickness, plate was glued along the edges of a rectangular window with dimensions 38×18 mm2 of a plexiglass holder (see Fig. 3). Thus, in this design, the lithium niobate plate was stretched over the holder window and both sides of the plate were mechanically free with maintaining the plane parallelism. On the underside of the plate, 2 interdigital transducers (IDTs) were pre-applied to excite and receive the acoustic wave. Each transducer had an aperture of 8 mm and contained 5 pairs of fingers with a period of 1.2 mm. The distance between the transducers was 27 mm.”

Page 5:

“between IDTs using the special holders. The gap between the container and delay line was provided with 8 μm thick aluminum foil strips. This design of the sensor allowed us to use a removable liquid container, which in turn facilitated cleaning the container from the sample under study and accelerated the analysis of many samples. The constant air gap between the plates ensured repeatability of the results, and also prevented the damage to the delicate delay line during cleaning the container from the spent biological sample. The thickness of the bottom of liquid container was equal 0.5 mm, the volume of the container was 1.5 ml.”

  1. The role of the narrow airgap between the two LiNbO3 sheets should also be explained. Why is this gap needed and what is done to assure that the tiny gap is kept constant over a relatively large area? It should also be emphasized how the wave energy is transferred between the sheets. I guess the electric field must play an important role here since the mechanical wave coupling through the airgap is quite weak?

Answer: We agree with the Reviewer and have added the following text on the page 5:

“Since the acoustic wave propagating in a thin Y-X LiNbO3 plate have a high value of the electromechanical coupling coefficient, the electric field accompanying this wave penetrates in the contacting air. Due to this property there exists the slot wave propagating in two piezoelectric plates separated by the vacuum (air) gap. The excitation of the slot wave in such structure leads to the appearance of the clearly expressed resonant peaks on the frequency dependence of the insertion loss of the output signal of the delay line. The appearance of the peaks is determined by the fact that the bottom of liquid sensor is limited in the direction of propagation of the acoustic wave. Each resonant peak corresponds to the case when the width of the second plate is equal to the whole number of the acoustic half-waves. The earlier studies [30] have shown that the depth of these peaks decreases with an increase in the gap between the piezoelectric plates. A gap of 30 μm is the limit for the existence of resonance peaks. Therefore, we have chosen the minimum gap of 8 μm, for which the resonance peaks are well pronounced.”

  1. The illustration of the experimental setup shown as the middle part of Fig. 3, should be improved by including physical dimensions. Dimensions and volume for the used fluid contained should also be included in the figure and/or manuscript.

We have pointed all dimensions of the sensor on the new texts on pages 4 and 5 (see answer on the comment 1).

  1. From what is said in the manuscript and in discussion, it’s rather difficult to identify the correlation between the observed peek shifts, and the contents of aqueous solutions. I would therefore recommend putting more emphasize on this in the discussion. Correlation to changes in the fluid conductivity is already mentioned, but changes in fluid wave properties (e.g. attenuation and phase velocity) might also be important here?

Answer: We agree with the Reviewer and therefore, on pages 10 and 11 and in the discussion, we once again emphasized the relationship between the change in the sensor output parameters and the specific interaction of viruses with antibodies. On pages 10 and 11, we added the following paragraph:

Thus, the interaction of viruses with specific antibodies leads to an increase in the conductivity of the suspension. This suspension is in contact with one of the plates of the structure with a piezoactive slot mode accompanied by an electric field in the suspension. A change in the conductivity of a suspension alters the attenuation and phase velocity of the slot mode. This, in turn, changes the depth and frequency of the resonance peaks in the frequency dependence of the insertion loss of the sensor. The depth and frequency of resonance peaks are directly measurable quantities and are therefore used as an analytical signal indicating the presence of viruses in suspension.

On the page 13 we have added the following paragraph:

“The interaction of viruses with specific antibodies leads to an increase in the conductivity of the suspension. A change in the conductivity of a suspension alters the attenuation and phase velocity of the slot mode. This, in turn, changes the depth and frequency of the resonance peaks in the frequency dependence of the insertion loss of the sensor.”

  1. It would also be interesting if the authors could discuss improvement of the current system with respect to for Coronaviruses detection. Perhaps there is something to gain from miniaturization the setup and increasing the frequency?

Answer: We have included the following text on the page 14:

“The presented set up showed for the first time the ability to quickly detect viruses in the aquatic environment. Without a doubt, it can be modernized and made more convenient for analysis. It consists of a sensor and an expensive S-parameter meter E5071C (Keysight Technologies, USA) (frequency range 9 kHz - 8.5 GHz). In the future, it is planned to make a cheaper S-parameter meter for the ~1-10 MHz range, combined with a computer with software for fast processing of results.”

Reviewer 2 Report

The authors are showing the first time measurement results for detection of Coronavirus by antibody biding with a SAW sensor device. Clearly the advantages of wide range of possible conductivities of aqueous solution together with the short detection time, makes the SAW sensor device very interesting for fast testing in real life environments. 

The novelty of the content is very high, the structure of the document very clear, the discussion of the results very percise and the reference good chosen. 

Only two minor questions I asked myself, after reading the paper and checking the references: 

  • How many times were the tests performed? What is the standard deviation, when the test are repeated several times?
  • Why is the summary of state of Coronavirus biosensors in the discussion chapter? For me it belongs to the state of science / technology in the introduction chapter. 

All together I strongly recommand to publish this paper in the Sensors Journal. 

Author Response

Answers to comments of the Reviewer 2

The authors are showing the first time measurement results for detection of Coronavirus by antibody biding with a SAW sensor device. Clearly the advantages of wide range of possible conductivities of aqueous solution together with the short detection time, makes the SAW sensor device very interesting for fast testing in real life environments. 

The novelty of the content is very high, the structure of the document very clear, the discussion of the results very percise and the reference good chosen. 

Only two minor questions I asked myself, after reading the paper and checking the references: 

  • How many times were the tests performed? What is the standard deviation, when the test are repeated several times?

Answer: We agree with the Reviewer 2 and we have included the following text on the page 6.

2.6. Statistical analysis

All experiments were performed in 5 replicates, and the final results were calculated from the averaged values. The average error of 5 measurements at each point of the frequency range did not exceed ± 2%. The results were statistically processed using standard procedures integrated into Excel 2007 (Microsoft Corp., USA). After the arithmetic mean and standard deviation were found for a given data sample, the standard error of the arithmetic mean and its confidence limits were determined taking into account the Student's coefficients (n, p) [number of measurements n = 5, probability = 95% (p = 0.05)].. Therefore, all the graphs stated below containing more than 1600 frequency points are given without the error bars, because in the presented scale they will simply be invisible.

  • Why is the summary of state of Coronavirus biosensors in the discussion chapter? For me it belongs to the state of science / technology in the introduction chapter

Answer: We have provided Table 3 in the “Discussion” in order to compare the proposed method with known ones. Comparisons were made in terms of analysis time and use / non-use of immobilized antibodies. For this we have included the analysis time and use / non-use of immobilized antibodies in the Table 3 and have corrected some sentences (marked by yellow).